# Development of a Prediction Model for Severe Hypoglycemia in Children and Adolescents with Type 1 Diabetes: The Epi-GLUREDIA Study

**DOI:** 10.3390/nu17162610

**Published:** 2025-08-12

**Authors:** Antoine Harvengt, Marie Bastin, Cédric Toussaint, Maude Beckers, Thibault Helleputte, Philippe Lysy

**Affiliations:** 1Pôle EDIN, Institut de Recherche Expérimentale et Clinique, UCLouvain, 1200 Brussels, Belgium; maude.beckers@saintluc.uclouvain.be (M.B.); philippe.lysy@saintluc.uclouvain.be (P.L.); 2DNAlytics, 1348 Ottignies-Louvain-la-Neuve, Belgium; marie.bastin@dnalytics.com (M.B.); cedric.toussaint@dnalytics.com (C.T.); thibault.helleputte@dnalytics.com (T.H.); 3Specialized Pediatrics Service, Cliniques Universitaires Saint-Luc, 1200 Brussels, Belgium

**Keywords:** severe hypoglycemia, type 1 diabetes, machine learning, continuous glucose monitoring, predictive modeling

## Abstract

**Background:** Severe hypoglycemia (SH) is a critical complication in children and adolescents with type 1 diabetes (T1D), associated with cognitive impairment, coma, and significant psychosocial burden. Despite advances in glucose monitoring, predicting SH remains challenging, as most models focus on milder hypoglycemic events. **Objective:** To develop a machine learning model for early prediction of SH using continuous glucose monitoring (CGM) data in children and adolescent T1D patients. **Methodology:** This retrospective study analyzed CGM data from 67 patients (37 SH episodes, 1430 non-SH segments). Glycemic curves were segmented into 5-day windows, and 21 features were extracted, including glycemic mean, variability, time below range (TBR < 60 mg/dL), and PCA components of glucose trends. A support vector machine (SVM) model was trained using repeated cross-validation to predict SH 15 min before onset. Model performance was evaluated using sensitivity, specificity, balanced classification rate (BCR), and area under the ROC curve (AUC). **Results:** The model achieved robust performance, with a median AUC of 90% (IQR: 87–93%) and median BCR of 84% (IQR: 80–89%). Sensitivity and specificity exceeded 80%, demonstrating reliable detection of impending SH. However, the positive predictive value (PPV) was low (12%), with false alarms frequently triggered during descending glucose trends or near-hypoglycemic values (end glucose <54 mg/dL). SH episodes were stratified into two subgroups: group 1 (<45 mg/dL, *n* = 26) and group 2 (>52 mg/dL, *n* = 15). Notably, false alarms occurred at a median interval of 25 days, minimizing alarm fatigue. **Conclusions:** These findings confirm the feasibility of SH prediction in clinical practice, prioritizing high-risk events over milder hypoglycemia. By alerting patients and medical teams early on, this tool could facilitate individualized treatment adjustments, reduce the risk of serious hypoglycemic events, and thus contribute to more personalized management of pediatric diabetes, while improving patients’ quality of life.

## 1. Introduction

Type 1 diabetes (T1D) is an autoimmune disorder that leads to repeated episodes of hyperglycemia. After initiation of insulin therapy, approximately 50–60% of patients with newly diagnosed T1D experience a temporary phase known as partial remission, characterized by a resurgence of residual β-cell insulin secretion. This phase presents clinically with a reduction in daily insulin needs and improved blood glucose control. Once partial remission ends, the reliance on exogenous insulin gradually increases, resulting in greater fluctuations in blood glucose levels and heightened challenges in preventing hypoglycemia [1].

Even with conventional intensive insulin therapy, frequent hypoglycemia (estimated at 5–15% of total blood glucose measurements) is often necessary to maintain the mean glycemic level within treatment targets (<150 mg/dL) [1]. Hypoglycemia is defined as a blood glucose level below 70 mg/dL and can manifest in various clinical forms. Among these, severe hypoglycemia (SH) is a critical event characterized by a low glycemia (<54 mg/dL) and severe cognitive impairment, including coma or convulsions, requiring third-party intervention to administer carbohydrates or glucagon. Its incidence is estimated to be between 10 and 15 episodes per 100 children and adolescents under the age of 15 per year [2].

SH has a significant psychosocial impact, disrupting patients’ daily lives, schooling, and participation in sports activities [3]. Several risk factors have been identified, including a high frequency of hypoglycemia, a history of SH, poor glycemic control, or alcohol consumption in adolescents [2].

In recent years, technological innovations such as insulin pump therapy and continuous glucose monitoring (CGM) systems have emerged to improve diabetes management, leading to a reduction in the risk of hypoglycemia, particularly severe hypoglycemia [2,4,5]. These devices have been complemented by advanced mathematical algorithms designed to anticipate glycemic fluctuations and provide preventive warnings to patients [6,7,8]. Predictive models based on statistical methods, machine learning, and deep learning have been developed. Among these, the most widely used algorithms for predicting acute complications of diabetes (hypoglycemia and hyperglycemia) include the Kalman filter and classification models (such as support vector machines, k-nearest neighbors, and random forests), which achieve accuracy levels ranging from 70% to 99% [9]. For instance, in 2024, C. Duckworth et al. developed a machine learning model capable of predicting hypoglycemia (<70 mg/dL) and hyperglycemia (>270 mg/dL) with an accuracy exceeding 95% [10]. However, these models primarily focus on predicting glycemic values rather than the occurrence of severe symptoms, which is a major source of stress for children and adolescents with diabetes.

Our GLUREDIA study aims to develop an innovative model specifically designed to predict SH in children and adolescents with T1D. Unlike existing approaches, our tool is designed to predict SH based on the temporal evolution of blood glucose levels rather than mild hypoglycemia, which is less consequential for the patient. With this tool, we hope to considerably improve the quality of life of our children and teenagers with T1D.

## 2. Methodology

### 2.1. Study Design and Participants

The Epi-GLUREDIA study, a subsidiary part of the GLUREDIA consortium study, is a retrospective, monocentric clinical study designed to characterize SH in children and adolescents with T1D. The study took place entirely in the pediatric diabetes department of Cliniques universitaires Saint-Luc (CUSL, Brussels, Belgium). All data were collected and analyzed in accordance with the ethical Declaration of Helsinki. The GLUREDIA study was approved by the Hospital-Faculty Ethics Committee of the CUSL (2022/02/FEV/043). Patient enrollment began on 1 July 2017 and ended on 30 June 2024. All participants provided informed consent for the use of their clinical data in the Epi-GLUREDIA study. All medical data were fully anonymized to ensure confidentiality and compliance with ethical and regulatory standards.

The inclusion and exclusion criteria for participation in this study have been detailed in a previous publication [11]. Briefly, eligible participants were children and adolescents aged 2 to 18 years with a diagnosis of T1D based on ADA criteria and the presence of at least one anti-islet autoantibody [1,12,13]. Key exclusion criteria included age outside the 2–18 year range, use of medications affecting insulin metabolism, recent diagnosis of celiac disease, other autoimmune or malignant diseases, class III obesity (BMI Z-score > +3 SD) [14], major organ insufficiencies, suspected genetic syndromes, or recent participation in another clinical trial involving immunomodulatory treatments.

### 2.2. Study Procedure

The Epi-GLUREDIA retrospective study consisted of an analysis of the clinical and glycemic parameters encoded in the medical records (EPIC^®^ software, © 1979–2024 Epic System Corporation, Verona, WI, USA) of patients currently enrolled in the CUSL pediatric diabetes convention of care. Clinical data (clinical and glycemic parameters) were recorded at each report of clinical follow-up visit of all our patients.

### 2.3. Definition of Severe Hypoglycemia

SH events were characterized by both an intense hypoglycemia (<54 mg/dL) accompanied by an alteration of consciousness, ranging from mild confusion to complete loss of consciousness, with or without seizure. The occurrence and timing of these events were either reported by the patient and their family or documented in the medical record when emergency medical care was involved.

### 2.4. Continuous Glucose Monitoring

Raw continuous glucose monitoring metrics from children and adolescents with T1D from various continuous glucose monitoring devices (i.e., Freestyle Libre^®^ (Abbott, Wavre, Belgium); Dexcom^®^ (Dexcom, Temse, Belgium); Enlite^TM^ (Medtronic MiniMed, Bruxelles, Belgium)). A total of 67 continuous glucose curves (CGM) were used to train and test different predictive models. Of these curves, 39 contained at least one SH defined as blood glucose below 54 mg/dL and loss of consciousness, and 28 did not. The characteristics of SH events are detailed in the paragraph above.

### 2.5. Data Pre-Processing

To ensure data quality prior to their use in the models, several pre-processing steps were carried out. Firstly, blood glucose measurements from different sources were combined to produce a single CGM curve per patient. Next, the curves were resampled at a uniform frequency of 15 min to standardize the analysis and facilitate comparisons. Short-duration data gaps were filled by linear interpolation and data-free periods longer than two hours were eliminated. Then, the remaining segments were divided into five-day fragments to standardize curve length. The segments containing an episode of SH were divided in the following way: first, the five days before the SH episode were removed from the rest of the segment, with the moment of the SH being the last point; then the rest of the segment was divided into periods of 5 days. For the segments containing no SH, the procedure was the same, except that the first step was ignored. After this processing, the curves were labeled and separated into two categories: those that ended in an SH event (37 curves) and those that did not (1430 curves) (Figure 1—flowchart).

Figure 2A,B depict an example of 5-day curves, one ending with an SH event one not. Each 5-day CGM curve was then characterized by a set of 21 features. These features included glycemic parameters such as glycemic mean, variability (mean/standard deviation ratio), the frequency of grade 1 hypoglycemia (glycemia < 60 mg/dL), the proportion of time spent affected by hypoglycemia (time below the range; TBR_<60_), the proportion of time spent affected by normoglycemia (time in range; TIR_60–160_), the proportion of time spent affected by hyperglycemia (time above the range; TAR_>160_), the first five components of a Principal Component Analysis (PCA) realized on the whole curves, the first five components of a PCA realized on the five last hours of every curves, a Boolean feature indicating if during the last 45 min the curve was increasing or decreasing, and another Boolean feature indicating if the last point of the curve was lower than 54 mg/dL. In addition to these 18 features, the age and sex of the patient and the age of the patient when they were diagnosed with diabetes were also included. Those 21 features will be used to predict if a 5-day CGM curve is ending with an SH event or not. The parameters used to develop the model were primarily selected based on preliminary findings from a previous study conducted by our team. Additional parameters were chosen based on clinical expertise and insights drawn from the existing scientific literature.

To investigate the capacity of the model to predict SH, the last point of each 5-day CGM curve was removed before computing all the features described above. This is equivalent to predicting 15 min in advance whether the curve will end with an SH event or not.

### 2.6. Univariate Analyses

Univariate analyses consist of bilateral hypothesis tests performed on the response (curve ending with SH or not) vs. each feature in the dataset. Depending on the nature of the second feature (numeric or binary), the statistical test used was either a Kruskal–Wallis test [15] or Fisher’s test [16]. The *p*-value of the statistical test was corrected for multiple testing following the Benjamini–Hochberg method [17].

### 2.7. Modeling Pipeline

To predict whether the curve will end up in SH or not, we built a multivariate predictive model evaluated in a cross-validation framework.

The following steps were repeated a large number of times (*k*):
○Randomly select 80% of the patients to form the training set;○Normalize features and store normalization parameters for next step;○Rank features using feature selection method *f*;○Learn model *m* using the top *s* features.For the 20% of the remaining patients (test set), the following steps were performed:
○Apply normalization using normalization parameters computed on the training set;○Predict SH or not on these unseen patients curves with model *m*;○Compute predictive performance indices by comparing predictions with true labels.Compute over the *k* resamplings:
○The average predictive performances;○Feature selection stability.

We also describe additional technical aspects of the modeling pipeline. Instead of learning one single model to extract the subset of markers, multiple models (k = 200) were learned on random subsamples, and a consensus selection was made. A robust normalization of the features was performed by subtracting the median value and dividing it by the interquartile range. Three feature selection methods f were tested: SVM-based algorithm [18], random forest [19] and a consensus of Wilcoxon tests. Two classification models m were tested: support vector machine (SVM), and random forest.

The predictive performances metrics used to evaluate the model are derived from a confusion matrix in which we report true positives (TP), false positives (FP), true negatives (TN), and false negatives (FN) as depicted in Table 1 below.

The metrics are as follows:

The sensitivity: $\frac{TP}{TP+FN}$

The specificity: $\frac{TN}{TN+FP}$

Positive Predictive Value (PPV): $\frac{TP}{TP+FP}$

Balanced Classification Rate (BCR): $\frac{sensitivity+specificity}{2}$

The Area Under the Receiver Operating Characteristic (ROC) Curve (AUC);

The BCR was preferred to accuracy due to the high-class imbalance (37 positive cases vs. 1430 negative cases). The stability metric which describes how much the *k* sets of *s* features have in common is evaluated with the Kuncheva index [20]. Several signature sizes *s* were evaluated. The cross-validation framework accounts for real-life use simulation by either including or excluding all curves for a patient in the training set. This way, the model is evaluated only on curves from patients that were not included during the training phase.

### 2.8. SHAP Values

A SHapley Additive exPlanations (SHAP) value [19] analysis was performed to better understand the behavior of the model. We looked at true positives, true negatives, and false positives. The SHAP values were computed with the R package fastshap (R-4.5.1). Each prediction was associated with its SHAP value waterfall graph which depicts the contribution of each predictor of the model for that prediction in particular.

## 3. Results

### 3.1. Characterization of SH Based on CGMs

The first step in our study was to numerically characterize SH events based on glycemic variations recorded by CGMs. Figure 2C illustrates each of the SH events experienced by our patients described in Table 1. The analysis took into account glycemic data within a time window defined by the initial glucose decline and the first glycemic peak following the SH event. Two distinct subgroups of SH were identified: group 1 (*n* = 26) comprised episodes where blood glucose fell below 45 mg/dL, while group 2 (*n* = 15) included episodes where blood glucose remained above 52 mg/dL. Before the start of hypoglycemia, no significant difference in terms of pre-hypoglycemia glycemic evolution was observed between the two groups. However, after the onset of the SH episode, notable differences emerged: the mean duration of hypoglycemia was significantly longer in group 1 than in group 2 (161.3 ± 118.9 vs. 106.1 ± 87.2 min, *p* < 0.05). In addition, the area under the curve during SH was significantly higher in group 1 than in group 2 (188.42 ± 151.1 vs. 53.8 ± 44.1 mg.min/dL, *p* < 0.001). Finally, peak blood glucose was higher in group 2 than in group 1 (53.0 ± 7.2 vs. 40.3 ± 10.3 mg/dL, *p* < 0.001).

### 3.2. PCA Decomposition

Figure 3 depicts the decomposition of the last 5 h of the 5-day glycemic curves into four principal components projected back in the original space and colored by the response (i.e., the presence or absence of SH). The decomposition into principal components showed a difference in amplitude and oscillations between positive and negative labels.

### 3.3. Results of Univariate Analyses

Univariate analyses did not provide any relevant results. The results can be found in Appendix A.

### 3.4. Predictive Model

The best predictive model obtained is an SVM using all 21 features described in Section 2—Data pre-processing. The model showed good overall predictive performance. Across the 200 resamplings, the median AUC reached 90% (IQR: 87–93%), and the median BCR was 84% (IQR: 80–89%), both suggesting consistently high model performance of around 85%. Both the median sensitivity and median specificity were superior to 80%. This means that the model was able to correctly identify at least eight positive cases out of ten and eight negative cases out of ten. The median PPV was equal to 12%, meaning that the model frequently predicted false positive cases of SH. Boxplots of the above metrics computed over the 200 resamplings are reported in Figure 4A. The ROC curve is shown in Figure 4B. It is currently premature to report metrics such as the Net Reclassification Index (NRI) or Decision Curve Analysis (DCA), as there are no existing models with comparable predictions and no fixed decision threshold established to balance sensitivity and specificity. These measures will be incorporated in future work once the model is further developed and clinically calibrated to support meaningful evaluation of clinical benefit.

### 3.5. SHAP Values

Figure 5 depicts representative examples of waterfall plots generated on a true positive, a true negative, and a false positive prediction. In any case, the feature contributing the most to the model decision is almost always the shape of the 5-day curve over the last five hours (gathering the four components of the PCA). Two other important features are the glycemic mean computed over the 5-day curve and whether the end of the 5-day curve is descending or not. To go a step further into the comprehension of false positive predictions, in Figure 6 we show the mean curve and 95% confidence interval on all 5-day curves resulting in a false positive prediction (glycemic curve of the past five days before the prediction time point). If we look at the end of the curve, i.e., the last hours, we can clearly see a major drop as well as a strong narrowing of the confidence interval. In the same fashion, the last glycemic value of each false positive prediction is rather low.

## 4. Discussion

Our study confirms the feasibility of developing a predictive model for SH in a setting of follow-up of children and adolescents with T1D. The model predictive performances were assessed in a cross-validation framework simulating real-life use, by putting all data curves of a patient either in the training set or in the test set. As seen in the Section 3, the model globally shows very good predictive performances, with AUC, BCR, sensitivity and specificity all being over 80%. While machine learning-based models, such as the XGBoost model described by Duckworth et al., sometimes achieve higher AUC values for hypoglycemia prediction, our model distinguishes itself by specifically targeting SH rather than all hypoglycemic events, including grade 1 hypoglycemia, which are less clinically significant [7,10]. This focus on SH is critical for clinical utility, as it ensures that the model prioritizes the detection of events that pose the greatest risk to patients [2].

A high sensitivity is very important for clinical use, since this means that the model detects more than eight out of ten SH events. This performance is comparable to, or even exceeds, that of other models reported in the literature, which typically achieve sensitivities ranging from 70% to 99% [9]. SHAP value analysis revealed that the two most influential features contributing to the model’s predictions are the glycemic trend during the five hours preceding the event and the glycemic mean over the same period. These findings indicate that the model relies heavily on short-term glycemic dynamics to assess the risk of SH.

However, this high sensitivity comes with a trade-off: the model’s PPV remains low, at only 12%. In practice, this means that seven times of out eight, the model will trigger an alarm even though the patient is not about to experience an SH event. To better understand why the model generates that many false positives, we looked more closely at each false positive prediction. We found that each false positive prediction matched with a decreasing glycemic curve and/or a low-ending glycemic level. This is consistent with the SHAP analysis, which highlighted that the glycemic evolution in the 5 h preceding the event is the most influential feature in the model’s decision-making. In cases of true positives, this evolution captures the critical descent towards SH, whereas in false positives, it still reflects a similar downward trend that does not culminate in SH but may still represent a grade 2 hypoglycemia without severe symptoms. As such, while the model incorrectly predicts an SH event in these cases, it still identifies a clinically relevant situation. Consequently, the false alarm may not be markedly problematic as it can prompt a preventive action, such as carbohydrate intake, which may help avoid further glycemic deterioration. This observation aligns with findings from Puhr et al., who noted that predictive hypoglycemia alerts are generally well-tolerated by users, as they help prevent severe episodes, even if some alarms are triggered prematurely [5].

By detecting over 80% of SH episodes, our model has the potential to significantly reduce the risk of severe complications associated with hypoglycemia, such as hypoglycemic coma or even death. Additionally, the model could improve patients’ quality of life by alleviating anxiety related to hypoglycemia and enabling better self-management of diabetes [2]. Indeed, SH is often experienced as a highly stressful event, and the fear it induces can sometimes paralyze patients in their daily diabetes management. By providing a tool capable of alerting patients to the potential risk of severe hypoglycemia, the model may help reduce psychological barriers, empower patients to act proactively, and ultimately support more confident and effective disease management [2,3,21].

Finally, we looked more closely at the frequency of false alarms per patient. To simulate real-life use of the model, we evaluated it on each patient by leaving that patient out during training and applying the model on all glycemic curves. We then identified all false alarms (false positive predictions) and computed the mean delay in days between each false alarm for each patient. We can see that the median mean delay is equal to 25 days. So, even though the model triggers a significant amount of false positives, those are sufficiently spaced through time not to overwhelm the patient with false alarms. This reduces the risk of “alarm fatigue,” a common issue with continuous glucose monitoring (CGM) systems [7,10,22].

Despite its strong performance, our model could be further improved by incorporating additional data, such as physical activity, insulin doses, and carbohydrate intake, as suggested in previous studies [9,10,23]. Integrating these factors could reduce false positives and enhance the model’s precision. One limitation of this study is that the algorithm was developed and tested using data from a single clinical center and a relatively small cohort, which may restrict the generalizability of the results to broader populations. To address this, future studies should aim to validate the algorithm in large, multicenter cohorts that include more diverse patient profiles, and assess its performance in real-world clinical settings. If these results are confirmed, the next step would be to integrate the algorithm into medical devices such as glucose meters or smartphone applications connected to CGM systems which could provide real-time prediction and prevention of severe hypoglycemic events.

## 5. Conclusions

Our study confirms the feasibility of developing a clinically useful predictive model for SH in pediatric T1D patients, with robust performance (AUC > 90%, sensitivity/specificity >80%) in cross-validation simulating real-world use. While the model’s high false positive rate (PPV 12%) reflects its conservative approach to SH detection, these alerts often correspond to clinically relevant glycemic declines rather than benign fluctuations. Importantly, false alarms were spaced at a median interval of 25 days, minimizing alarm fatigue while maintaining vigilance for high-risk events.

### Strengths and Weaknesses

The main strength of our study lies in the development of an algorithm specifically designed to predict SH in patients with T1D. Unlike most existing models, which generate alerts for all hypoglycemic episodes, whether mild or severe, our algorithm focuses on identifying rarer but more dangerous events. Despite the low incidence of SH, the algorithm demonstrated high sensitivity, specificity, and AUC, highlighting its strong predictive performance. Another notable strength is the use of a pediatric and adolescent cohort, whereas most previous studies have focused on adult populations.

The main weakness of our study is the small sample size, derived from a single clinical center, which may restrict the generalizability of the findings. A second limitation is the relatively low positive predictive value. However, it is important to note that most false alarms correspond to upcoming episodes of significantly low glycemia, which still represent clinically relevant events that justify early warning.

## Figures and Tables

**Figure 1 nutrients-17-02610-f001:**
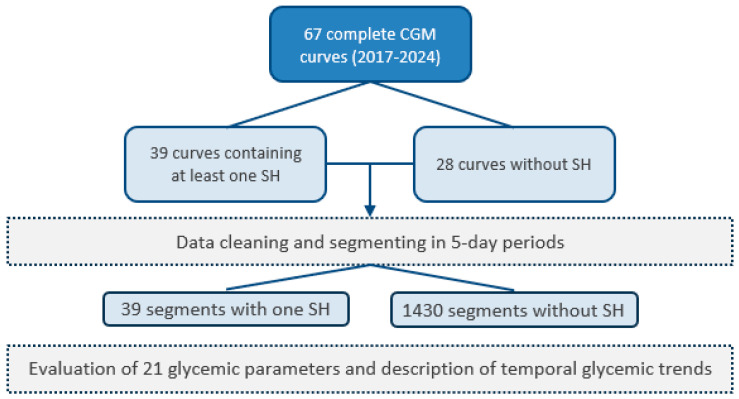
Study flowchart.

**Figure 2 nutrients-17-02610-f002:**
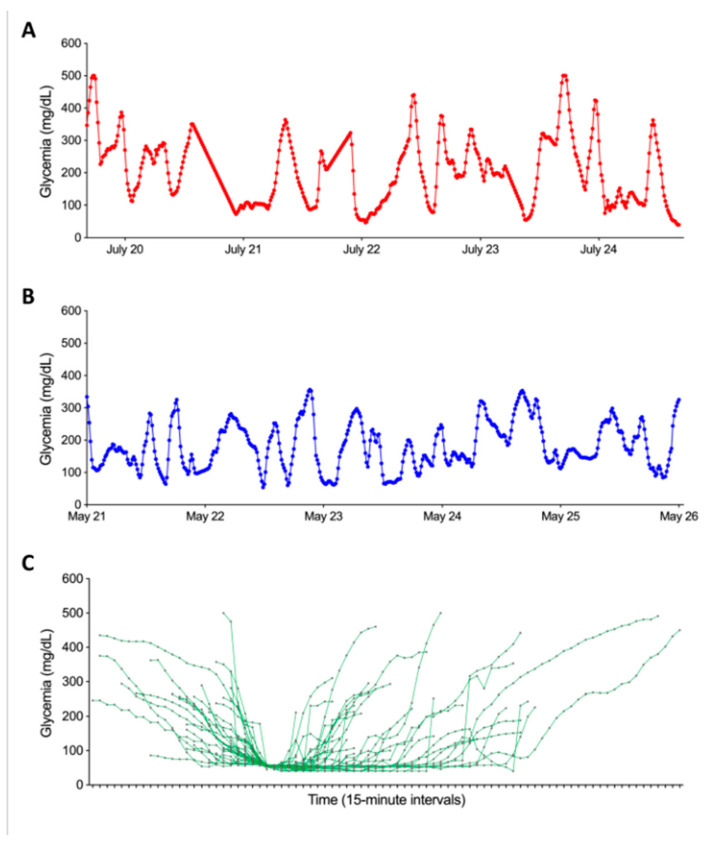
Graphical representation of the temporal evolution of glycemic curves in our dataset. Panel (**A**) illustrates the progression of glycemia during the five days ending with an SH event. Panel (**B**) presents the evolution of glycemia in the five days not ending with an SH event. Both panels share axes: *x*-axis = time (5-day window), *y*-axis = glucose (mg/dL). Panel (**C**) shows the graphical representation of the 39 SH based on CGM data synchronized to hypoglycemia onset (time 0). The *X*-axis represents the time relative to the onset of hypoglycemia, while the *Y*-axis shows glycemia in mg/dL. Each curve corresponds to a SH. The start of the curve marks the beginning of the glycemia drop, and the end corresponds to the first glycemic peak following the episode.

**Figure 3 nutrients-17-02610-f003:**
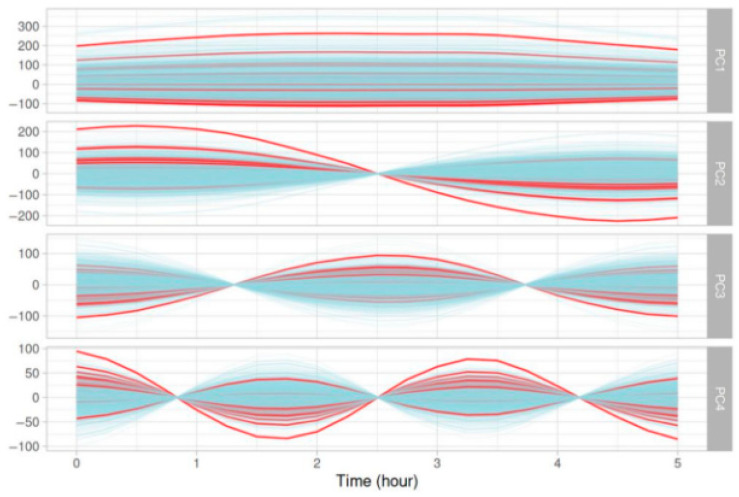
Graphical representation of the four principal components from a PCA analyzing the evolution of glycemic parameters over the last five hours of five-day glycemic segments. Red curves represent glycemic evolution ending with an SH event, while blue curves correspond to glycemic evolution not ending with an SH event.

**Figure 4 nutrients-17-02610-f004:**
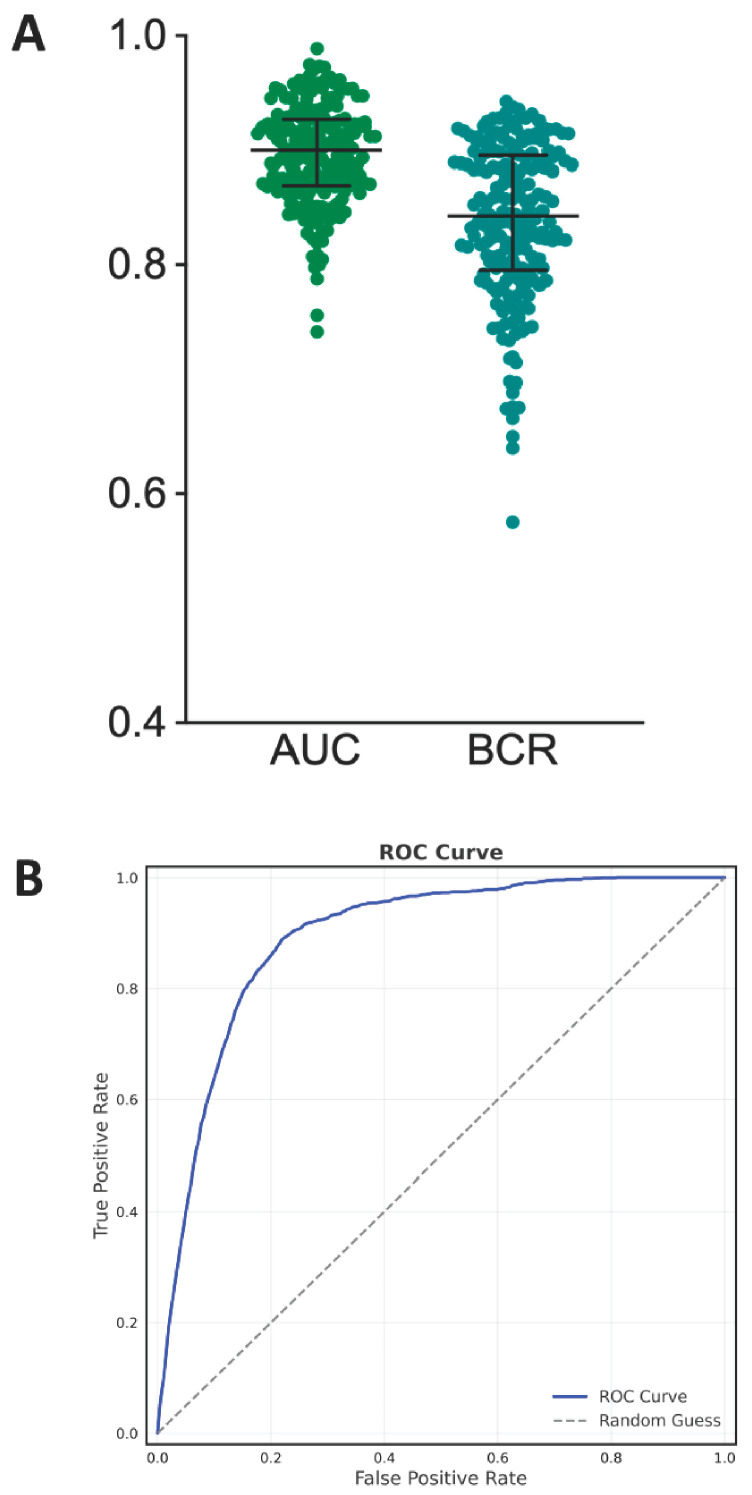
Panel (**A**) displays boxplots summarizing model performance across 200 resampling iterations, with AUC (median: 90%) and BCR (median: 84%) demonstrating consistent predictive accuracy. Panel (**B**) presents the ROC curve, where the *x*-axis showing false positive rate (1—specificity) and *y*-axis indicating true positive rate (sensitivity).

**Figure 5 nutrients-17-02610-f005:**
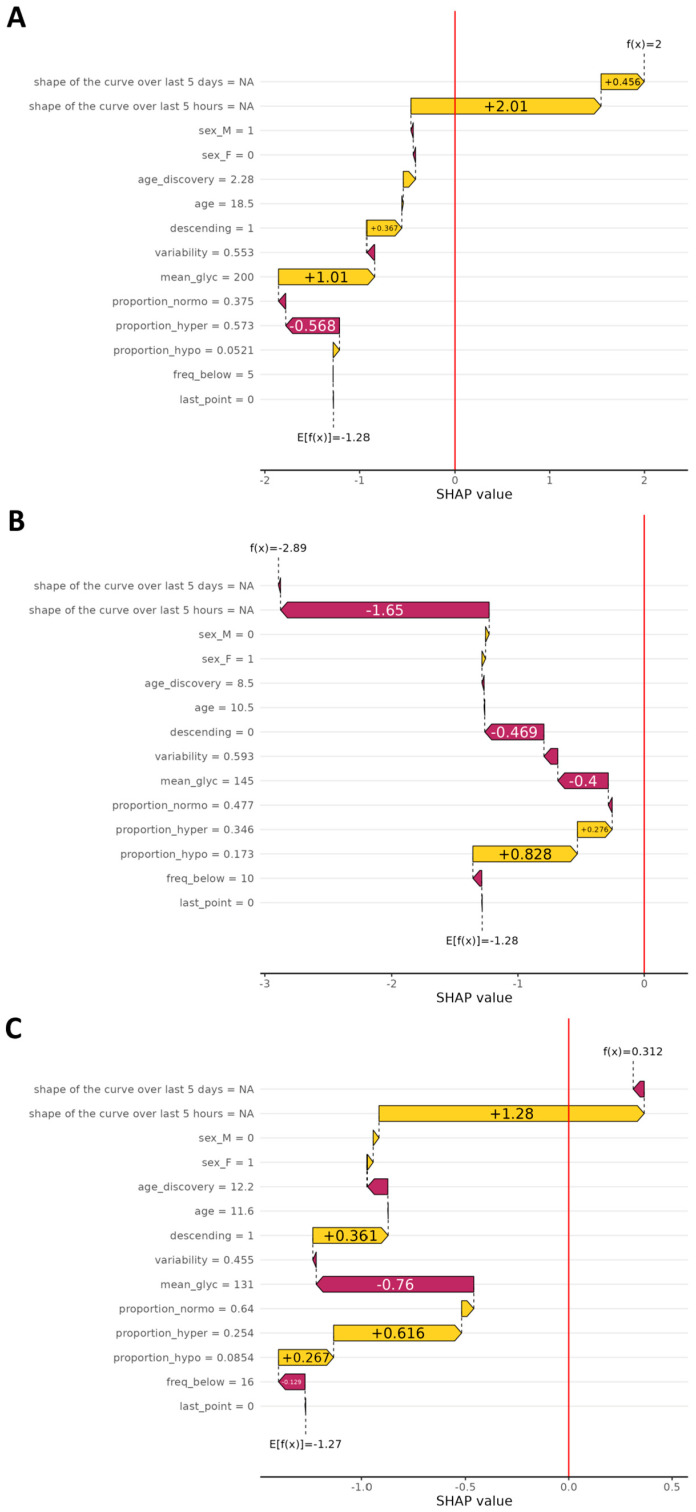
Waterfall plots illustrate the individual impact of each parameter on the final prediction of the SH model, based on SHAP values. In the upper part, **Panel A**, the plot shows contributions for true positive predictions (correctly identified SH). In the middle part, **Panel B**, the plot corresponds to true negatives (correct predictions of no SH). In the lower part, **Panel C**, it shows false positives (predicted SH event that did not occur). The *X*-axis represents the SHAP value, indicating how much each parameter contributes (positively or negatively) to the model’s decision for a given prediction. The *Y*-axis lists the glycemic parameters used in the model. A positive SHAP value drives the prediction toward SH, while a negative value shifts it away from that outcome.

**Figure 6 nutrients-17-02610-f006:**
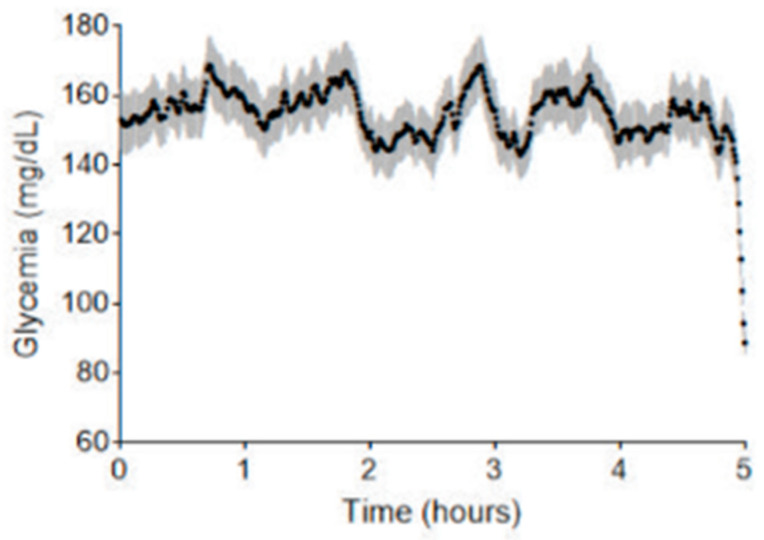
Temporal evolution of blood glucose levels for segments falsely identified as SH by the prediction model. The *X*-axis shows the five-day period of the analyzed segments, while the *Y*-axis represents blood glucose levels in mg/dL. This plot compiles all five-day segments that were falsely predicted to end in SH (false positives). Each point reflects the average blood glucose value at a given time across all such curves. The gray-shaded area around each point indicates the 95% confidence interval, representing the variability of glucose levels at that specific time.

**Table 1 nutrients-17-02610-t001:** Demographic profile and glycemic data of participants.

Cohort Description	
	T1D with SH (*n* = 41)
**Demographic Data**	
Gender—girl (%)	19/41 (46.34)
Age—years	13.32 ± 3.65
Puberty	
Pre-puberty—*n* (%)	11/41 (26.83)
Puberty—*n* (%)	30/41 (73.17)
**Clinical Data**	
Height—percentile	55.0 ± 25.5
Weight—percentile	61.0 ± 29.8
BMI—percentile	59.0 ± 27.9
**Glycemic Data**	
Diabetes duration—years	6.33 ± 4.60
Total daily insulin dose—UI/kg/day	0.88 ± 0.23
Medical treatment	
Pump—*n* (%)	6/41 (14.63)
2 injections—*n* (%)	8/41 (19.51)
Basal prandial—*n* (%)	27/41 (65.85)
HbA_1C_—%	7.43 ± 1.13
Glycemic mean—mg/dL	170.91 ± 33.34
TAR_>180_—%	44.24 ± 15.04
TIR_70–180_—%	44.72 ± 12.96
TBR_<70_—%	11.03 ± 6.59
Treatment of SH	
None—*n* (%)	2/41 (4.88)
Glucose—*n* (%)	27/41 (65.85)
Glucagon—*n* (%)	9/41 (21.95)
Emergency—*n* (%)	3/41 (7.31)

Plus-minus values are means ± SD. Percentages may not total 100 due to rounding. SH, severe hypoglycemia; TAR_>180_, time above the range (>180 mg/dL); TIR_70–180_, time in range (70–180 mg/dL); TBR_<70_, time below the range (<70 mg/dL).

## Data Availability

The original contributions presented in this study are included in the article/Appendix A. Further inquiries can be directed to the corresponding author.

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
