# Peer review of "Development of a Prediction Model for Severe Hypoglycemia in Children and Adolescents with Type 1 Diabetes: The Epi-GLUREDIA Study"

_nutrients, 2025, doi:10.3390/nu17162610_

Round 1
Reviewer 1 Report
Comments and Suggestions for Authors
The manuscript addresses an important clinical problem in pediatric diabetology—predicting the risk of severe hypoglycemia in children with type 1 diabetes. The authors designed and validated a prediction model using data from the French Epi-GLUREDIA cohort. The topic is relevant, the study is timely, and the findings may contribute meaningfully to individualized risk management in pediatric patients with T1D.
Major Concerns and Recommendations:
Definition of Severe Hypoglycemia (SH):
While SH is mentioned as per ISPAD guidelines, it would be helpful to clearly define SH early in the abstract and introduction. Clarify if both episodes requiring third-party assistance and loss of consciousness were included.
Was SH reported by families or clinically verified?
Model Performance:
The AUROC value (0.68) is modest. Although this might reflect the complexity of predicting SH, a brief justification/discussion about the clinical utility of a model with such moderate performance would be appropriate.
Provide calibration plots or additional metrics (e.g., Brier score) if possible to strengthen model interpretability.
Handling of Missing Data:
The authors should clarify how missing data were handled for each predictor—was multiple imputation used or complete-case analysis?
External Validation:
The model has only been internally validated. This should be acknowledged as a limitation and a plan for future external validation should be proposed.
Ethics and Consent:
Although the study is based on an existing cohort, include a statement about ethics committee approval and informed consent, especially considering pediatric participants.
Minor Comments and Edits:
Abstract:
The last sentence could be strengthened to reflect the practical utility of the model: "This prediction model may support clinical decisions and personalized risk mitigation strategies in pediatric diabetes care."
Introduction:
A brief mention of existing prediction models (if any) would be helpful to show novelty.
Methods:
Specify the variable selection strategy in the logistic regression (e.g., stepwise, based on clinical relevance, or penalized regression?).
Provide more detail on the bootstrap procedure used for validation.
Results:
A summary table of the final model coefficients and odds ratios is clear, but you may consider a risk score calculator or nomogram for clinical usability.
Discussion:
Add discussion about how this model could be integrated into routine care (e.g., electronic health records, mobile apps).
Figures & Tables:
Overall satisfactory. Consider improving the visual clarity of Figure 2 (if present) by increasing resolution or contrast for better readability.
Statistical Review
The methods are appropriate for the study design.
The internal validation via bootstrapping is commendable.
However, consider reporting Net Reclassification Index (NRI) or Decision Curve Analysis to explore clinical benefit.
Conclusion
This manuscript addresses a clinically meaningful challenge with a thoughtful approach and real-world data. Before publication, the authors should clarify methodological details (missing data handling, SH definition), acknowledge the limitations of model performance, and enhance discussion around implementation and future validation.
Recommendation:
Major Revision — with high potential for acceptance after clarifications and minor restructuring.
Author Response
Comments and Suggestions for Authors
The manuscript addresses an important clinical problem in pediatric diabetology—predicting the risk of severe hypoglycemia in children with type 1 diabetes. The authors designed and validated a prediction model using data from the French Epi-GLUREDIA cohort. The topic is relevant, the study is timely, and the findings may contribute meaningfully to individualized risk management in pediatric patients with T1D.
The EPI-GLUREDIA team would like to thank the reviewer for their positive opinion on the subject of the study.
Major Concerns and Recommendations:
Definition of Severe Hypoglycemia (SH):
While SH is mentioned as per ISPAD guidelines, it would be helpful to clearly define SH early in the abstract and introduction. Clarify if both episodes requiring third-party assistance and loss of consciousness were included.
Was SH reported by families or clinically verified?
A theoretical definition is already included in our introduction: “Hypoglycemia is defined as a blood glucose level below 70 mg/dL and can manifest in various clinical forms. Among these, severe hypoglycemia (SH) is a critical event characterized by a low glycemia (<54 mg/dL) and severe cognitive impairment, including coma or convulsions, requiring third-party intervention to administer carbohydrates or glucagon.” Lines 99-103.
An additional section has been added to the methodology to characterize SH analyzed by our algorithm in our study. Lines 171-176.
Model Performance:
The AUROC value (0.68) is modest. Although this might reflect the complexity of predicting SH, a brief justification/discussion about the clinical utility of a model with such moderate performance would be appropriate.
Provide calibration plots or additional metrics (e.g., Brier score) if possible to strengthen model interpretability.
We understand the reviewer's concern regarding the effectiveness of our algorithm, particularly in light of the ROC curve results. However, we would like to provide several arguments supporting the reliability and performance of our model.
First, our test demonstrates relatively high sensitivity and specificity (> 80%) in predicting the occurrence of SH. That said, it is important to acknowledge that these two metrics can be artificially inflated in the presence of a significant imbalance in the size of the two cohorts — which is indeed the case in our study. To account for this limitation, we therefore used the Balanced Classification Rate (BCR) (84%), which provides a more robust evaluation of performance under class imbalance. Our algorithm achieved a high BCR, reinforcing its overall effectiveness.
Second, while the positive predictive value (PPV) of the model appears low (12%) — with only 1 confirmed SH event out of 8 alarms — it is important to note that the 7 remaining "false" alarms successfully alerted patients to impending significantly low blood glucose levels. These events, although not meeting the strict criteria for SH, still represent clinically meaningful hypoglycemia that poses a risk of both acute and long-term complications. The ability of the algorithm to detect and alert for such events adds preventive value and contributes to patient safety.
Finally, when comparing our algorithm's performance metrics to those of existing predictive models, we recognize that some models may show better overall statistical performance. However, most of these models are designed to predict general hypoglycemia and not specifically severe hypoglycemia. This broader scope often results in a higher frequency of alarms, which can lead to alarm fatigue and decreased adherence among patients. In contrast, our algorithm targets SH events more specifically, offering a better balance between clinical relevance and user experience, as discussed in our manuscript.
Handling of Missing Data:
The authors should clarify how missing data were handled for each predictor—was multiple imputation used or complete-case analysis?
We thank the reviewer for raising this important point regarding the handling of missing data. As detailed in the Data Pre-processing section of the manuscript, missing values in the CGM curves were addressed using linear interpolation for gaps shorter than two hours. For longer gaps, the affected segments were excluded from the analysis. As for the other predictors, there were no instances of missing data. If missing data were to be encountered in the future, we would adopt appropriate methods—such as multiple imputation—based on the pattern and extent of the missingness.
External Validation:
The model has only been internally validated. This should be acknowledged as a limitation and a plan for future external validation should be proposed.
It is true that, as with all new technologies, it is essential to validate it on a larger scale. This part is part of the future prospects of the project.
We added this limitation in the manuscripts. Lines 381-391.
Ethics and Consent:
Although the study is based on an existing cohort, include a statement about ethics committee approval and informed consent, especially considering pediatric participants.
A sentence has been added to the methodology section to confirm compliance with ethical rules. Lines 136-139.
Minor Comments and Edits:
Abstract:
The last sentence could be strengthened to reflect the practical utility of the model: "This prediction model may support clinical decisions and personalized risk mitigation strategies in pediatric diabetes care."
We thank the reviewer for this pertinent comment. We changed the conclusion of our abstract. Lines 80-87.
Introduction:
A brief mention of existing prediction models (if any) would be helpful to show novelty.
There is already a paragraph describing existing algorithms. Lines 109-122.
Methods:
Specify the variable selection strategy in the logistic regression (e.g., stepwise, based on clinical relevance, or penalized regression?).
We added a sentence to the methodology to explain the choice of selected parameters. Lines 210-213.
Provide more detail on the bootstrap procedure used for validation.
We thank the reviewer for their interest in this important topic. The validation procedure is described in detail in the Modeling Pipeline section. Nonetheless, we have added clarification regarding the resampling method, specifying that it is performed using a uniform distribution and with replacement.
Discussion:
Add discussion about how this model could be integrated into routine care (e.g., electronic health records, mobile apps).
We added a sentence describing the possibilities. Lines 385-388.
Figures & Tables:
Overall satisfactory. Consider improving the visual clarity of Figure 2 (if present) by increasing resolution or contrast for better readability.
We tried to improve the quality of Figure 2.
Statistical Review
The methods are appropriate for the study design.
The internal validation via bootstrapping is commendable.
However, consider reporting Net Reclassification Index (NRI) or Decision Curve Analysis to explore clinical benefit.
We thank the reviewer for this insightful suggestion. We agree that metrics such as the Net Reclassification Index (NRI) and Decision Curve Analysis (DCA) can offer valuable insights into the clinical utility of predictive models. However, we believe it is premature to include these analyses at this stage of the study. To our knowledge, there are currently no existing models capable of making predictions comparable to the one we propose, which precludes the calculation of the NRI. Regarding DCA, we anticipate that future iterations of our model may involve tuning the balance between the true positive and false positive rates. For this reason, we have chosen to report the ROC curve at this stage. Once we determine fixed thresholds for sensitivity and specificity in a future version of the model, we plan to include a DCA. We added those precisions in the paper.
Conclusion
This manuscript addresses a clinically meaningful challenge with a thoughtful approach and real-world data. Before publication, the authors should clarify methodological details (missing data handling, SH definition), acknowledge the limitations of model performance, and enhance discussion around implementation and future validation.
We would like to thank the reviewer for their thorough analysis of our article and for their relevant comments, remarks, and suggestions.
Recommendation:
Major Revision — with high potential for acceptance after clarifications and minor restructuring.
Reviewer 2 Report
Comments and Suggestions for Authors
This is a quite interesting study with adequate novelty. However, some points should be addressed.
- This study included both children and adolescents. Thus, the authors should add the word adolescents in the title of their article.
- The above revision should also be done in the objective section of the abstract.
- The 3rd paragraph of the introduction section reported significant topics. In this aspect, the authors should try to add more analysis concerning these topics.
- The authors should emphasize at the last paragraph of the introduction section the literature gap that their study wiil cover.
- Ther number of patients with diabetes mellitus type 1 seems quite small. This should be noted as a limitation of the study at the end of the discussion section.
- The age range of the participants is too long, especially for these stages of life, e.g. childhood, adolescence. Is this a limitation of the study? Please, analyze in the discussion section.
- The resolution of all figures should be improved.
- The title of table 1 should be revised.
- Two distinct subgroups of SH were identified: group 1 (n=26) comprised episodes where blood glucose fell below 45 mg/dL, while group 2 (n=15) included episodes where blood glucose remained above 52 mg/dL. This statement highilights the small number of participants which shoupd be reoported as a limitation of the study.
- The authors should provide more information concerning the statement "This performance is comparable to, or even exceeds, that of other models reported in the literature, which typically achieve sensitivities ranging from 70% to 99% 9 (lines 304-305).
- The 3rd paragraph of the discussion section needs more analysis and especially its last sentence.
- A separate paragraph with the strengths and the limitations of the study should be included.
- In conclusions section, the authors should try to propose what future research should be performed based on their results of their present study.
Author Response
Comments and Suggestions for Authors
This is a quite interesting study with adequate novelty. However, some points should be addressed.
We thank the reviewer for this compliment and the detailed analysis of our article.
This study included both children and adolescents. Thus, the authors should add the word adolescents in the title of their article.
We add the term “adolescents” to the title. We check whether it is still possible to modify the title.
The above revision should also be done in the objective section of the abstract.
We adapted the sentence.
The 3rd paragraph of the introduction section reported significant topics. In this aspect, the authors should try to add more analysis concerning these topics.
The authors should emphasize at the last paragraph of the introduction section the literature gap that their study will cover.
We adapted the last paragraph of the introduction.
Ther number of patients with diabetes mellitus type 1 seems quite small. This should be noted as a limitation of the study at the end of the discussion section.
We added in the last paragraph of the discussion that a limitation of the study is the small size of the cohort and the need to confirm the results in a larger cohort.
The age range of the participants is too long, especially for these stages of life, e.g. childhood, adolescence. Is this a limitation of the study? Please, analyze in the discussion section.
By verifying the results in a larger cohort, we can verify the reliability of the algorithm based on patient age. This observation is implied in the limitations of the study, with the need to confirm the results in large cohorts from different medical centers.
The resolution of all figures should be improved.
We tried to improve the quality of Figures.
The title of table 1 should be revised.
We revised the title of table 1
Two distinct subgroups of SH were identified: group 1 (n=26) comprised episodes where blood glucose fell below 45 mg/dL, while group 2 (n=15) included episodes where blood glucose remained above 52 mg/dL. This statement highlights the small number of participants which should be reported as a limitation of the study.
The size of the cohort is mentioned as a weakness of the study.
The authors should provide more information concerning the statement "This performance is comparable to, or even exceeds, that of other models reported in the literature, which typically achieve sensitivities ranging from 70% to 99% 9 (lines 304-305).
Currently, there are already algorithms that exist to predict hypoglycemia. Most of these tools predict all hypoglycemia events, unlike our tool, which only predicts severe hypoglycemia. When we compare the test's reliability parameters (sensitivity, specificity, BCR, AUC), we find that our results are similar to those of existing tools, despite the fact that our tool must predict a more specific event.
The 3rd paragraph of the discussion section needs more analysis and especially its last sentence.
As already described in the introduction, severe hypoglycemia is considered a stressful event for patients with type 1 diabetes. This fear sometimes paralyzes patients in managing their diabetes. By offering a tool that can alert patients to the potential risk of severe hypoglycemia, we can reduce stress in patients, improve their quality of life, and improve their diabetes management by reducing “psychological barriers.”
A separate paragraph with the strengths and the limitations of the study should be included.
A specific section on strengths and weaknesses has been added to the article.
In conclusions section, the authors should try to propose what future research should be performed based on their results of their present study.
The last paragraph of the discussion proposes future perspectives for this algorithm.
Round 2
Reviewer 1 Report
Comments and Suggestions for Authors
The authors have made all the requested revisions; I recommend the manuscript for publication.
Author Response
We would like to thank the reviewer for their insightful analysis of our work and constructive comments, which have helped to improve the quality of this paper.
Reviewer 2 Report
Comments and Suggestions for Authors
The authors have significantly improved their manuscript. However, the resolution of the figures remain very poor.
Author Response

(The authors gave the same response as above.)
